# Leading basic modes of spontaneous activity drive individual functional connectivity organization in the resting human brain

Xi Chen[1], Haoda Ren[1], Zhonghua Tang[1], Ke Zhou [2], Liqin Zhou[2], Zhentao Zuo[3], Xiaohua Cui[1], Xiaosong Chen [1], Zonghua Liu[4], Yong He [5,6,7,8] & Xuhong Liao [1,6✉]

Spontaneous activity of the human brain provides a window to explore intrinsic principles of functional organization. However, most studies have focused on interregional functional connectivity. The principles underlying rich repertoires of instantaneous activity remain largely unknown. We apply a recently proposed eigen-microstate analysis to three resting-state functional MRI datasets to identify basic modes that represent fundamental activity patterns that coexist over time. We identify five leading basic modes that dominate activity fluctuations. Each mode exhibits a distinct functional system-dependent coactivation pattern and corresponds to specific cognitive profiles. In particular, the spatial pattern of the first leading basis mode shows the separation of activity between the default-mode and primary and attention regions. Based on theoretical modelling, we further reconstruct individual functional connectivity as the weighted superposition of coactivation patterns corresponding to these leading basic modes. Moreover, these leading basic modes capture sleep deprivation-induced changes in brain activity and interregional connectivity, primarily involving the default-mode and task-positive regions. Our findings reveal a dominant set of basic modes of spontaneous activity that reflect multiplexed interregional coordination and drive conventional functional connectivity, furthering the understanding of the functional significance of spontaneous brain activity.

[1] School of Systems Science, Beijing Normal University, Beijing 100875, China. [2] Beijing Key Laboratory of Applied Experimental Psychology, School of Psychology, Beijing Normal University, Beijing 100875, China. [3] State Key Laboratory of Brain and Cognitive Science, Institute of Biophysics, Chinese Academy of Sciences, Beijing 100101, China. [4] State Key Laboratory of Precision Spectroscopy, School of Physics and Electronic Science, East China Normal University, Shanghai 200241, China. [5] State Key Laboratory of Cognitive Neuroscience and Learning, Beijing Normal University, Beijing 100875, China. [6] Beijing Key Laboratory of Brain Imaging and Connectomics, Beijing Normal University, Beijing 100875, China. [7] IDG/McGovern Institute for Brain Research, Beijing Normal University, Beijing 100875, China. [8] Chinese Institute for Brain Research, Beijing 102206, China. ✉email: liaoxuhong@bnu.edu.cn

Spontaneous activity in the resting human brain exhibits well-organized spatiotemporal patterns, providing a window into understanding the intrinsic functional organization[1,2]. Using resting-state functional magnetic resonance imaging (R-fMRI), numerous studies have revealed the large-scale functional connectivity (FC) network by measuring low-frequency spontaneous fluctuations of blood-oxygenation-level-dependent (BOLD) signals[3–5]. The functional network exhibits non-trivial properties, such as functionally specific but interacting modules[6–8], which facilitate efficient functional segregation and integration across the brain[9–11]. Furthermore, the functional network architecture varies across individuals[12–15], shapes functional activation patterns during tasks[16–19], is related to individual cognitive performance[18,20,21], and is modulated by the mental states[22,23].

Despite the success of the functional network analyses, the associated insights are limited to the connectivity patterns summarized over time. Accumulating evidence suggests that the interregional functional interaction is highly dynamic with time-varying patterns[24–26]. An innovative approach is to examine single frames of brain activity to reveal the transient coordination at shorter time scales (e.g., seconds)[27]. The whole-brain activity patterns have been classified into several recurrent brain states with different coactivation patterns[28–31]. The temporal transition between these brain states follows a hierarchical structure[31] and shows alterations across tasks[30,32,33], consciousness states[34,35], and psychiatric disorders[36,37]. In addition to the group-level analysis, a very recent study has identified individualized brain coactivation states, the occurrence rates of which depend on task states, handedness, and gender, and shows longitudinal changes in post-stroke recovery[38]. Although these studies provide valuable insights into the time-varying functional organization, they typically assign the instantaneous activity pattern at each time point to a single brain state; the commonality shared across time points has been underestimated[39]. A more natural view holds that multiple basic modes may coexist across the time-resolved activity, which are selectively combined at each time point to support potential cognitive responses[26,40,41]. Identifying these basic modes can unravel the building blocks of intrinsic activity and may provide an avenue to explore the multiplicity of the interregional relationships at rest. However, the spatial patterns of these basic activity modes and their potential functional significance remain largely unknown.

Recent R-fMRI studies have attempted to bridge the gap between instantaneous brain activity and FC patterns. For example, the point process analysis shows that FC profiles for regions of interest can be inferred from interregional coactivation patterns at specific time points[28,42]. Similarly, the edge-centric approach decomposes FC into framewise contributions[43] and reveals dominant contributions of high-amplitude coactivations at critical time points[44,45]. A recent study further reports that interregional FC relies on all time points, even those with low amplitudes[46]. Thus, we hypothesize that the basic modes of time-solved activity may make a substantial contribution to the FC pattern.

To address these issues, we leveraged a recently proposed statistical physics approach, i.e., the eigen-microstate analysis[47,48], to identify basic modes of spontaneous activity of the resting human brain. The eigen-microstate analysis is useful for extracting meaningful and fundamental spatial components (i.e., basic modes) underlying the temporal evolution of complex systems by incorporating spatial information over time[47,48]. Specifically, we applied this approach to R-fMRI data from healthy young adults from three datasets: the S900 release of the Human Connectome Project (HCP)[49], the sleep-deprivation dataset[50], and the Beijing Zang dataset[51]. First, we identified

the leading basic modes that dominated the spontaneous fluctuations of BOLD signals and unraveled their cognitive significance. Second, we developed a theoretical model to elucidate how these basic modes contribute to the whole-brain FC pattern and verified this model by empirically reconstructing the FC pattern. Finally, we investigated whether these basic modes are affected by the modulation of mental states, e.g., by sleep deprivation.

## Results

**A small number of basic modes dominated spontaneous activity.** We employed two runs (i.e., REST1 and REST2) of R-fMRI data from 700 participants selected from the HCP dataset and extracted regional time courses for 1000 cortical nodes based on a prior functional parcellation[52]. Then, we identified the basic modes at the population level for each run by applying the eigen-microstate analysis[47] to the concatenated time courses across participants (Fig. 1). For both runs, the weights of the basic modes decreased rapidly with increasing ranking and reached an elbow point at the 6th basic mode (Fig. 1a). The first five basic modes before the elbow point accounted for a large proportion of the variance in activity (29% for REST1 and 28% for REST2) (both $ps < 0.001$, 10,000 permutations) and hereafter are referred to as the leading basic modes.

Each leading basic mode showed a heterogeneous spatial pattern, representing a typical activity mode underlying the rich repertoire of spontaneous activity (Fig. 1b); in the figure, opposite signs in the amplitude indicate opposite phases in the temporal fluctuation. The spatial patterns of these modes were highly similar between two runs (i.e., REST1 and REST2) (Supplementary Fig. 1). Based on a prior brain parcellation with seven functional systems[53], we found that the spatial patterns of the leading basic modes were system-dependent (Fig. 1b, c). For basic mode 1, positive amplitudes were mainly located in regions of the default-mode and frontoparietal networks, whereas negative amplitudes were mainly located in regions of the somatomotor and visual networks, as well as those of the ventral and dorsal attention networks. This pattern is similar to the previously reported principal gradient of FC[54], suggesting a hierarchical separation of brain activity between transmodal regions and primary and attentional regions. For basic mode 2, positive amplitudes were mainly located in regions of the default-mode, somatomotor, and visual networks, while negative amplitudes were mainly located in the regions of the frontoparietal and ventral/dorsal attention networks. For basic mode 3, positive amplitudes were primarily located in regions of the somatomotor, ventral attention, frontoparietal, and lateral default-mode networks, while negative amplitudes were primarily located in the visual and dorsal attention networks. For basic mode 4, positive amplitudes were mainly located in regions of the frontoparietal and dorsal attention networks, whereas negative amplitudes were mainly located in the ventral attention and medial visual networks. Basic mode 5 showed a finer spatial structure, in which heterogeneous amplitudes were observed within each functional system, and positive and negative amplitudes were mainly located in lateral and medial default-mode regions, respectively. These results suggest that a small set of basic modes govern the spontaneous fluctuations of brain activity, each of which shows a distinct coactivation pattern between functional systems.

We further identified the basic activity modes for each run at the individual level. We observed a small number of leading basic modes for most of the participants (number of leading basic modes, range: 3–10; mean ± std = 5.40 ± 1.39 for REST1 and mean ± std = 5.46 ± 1.39 for REST2) (Supplementary Fig. 2).

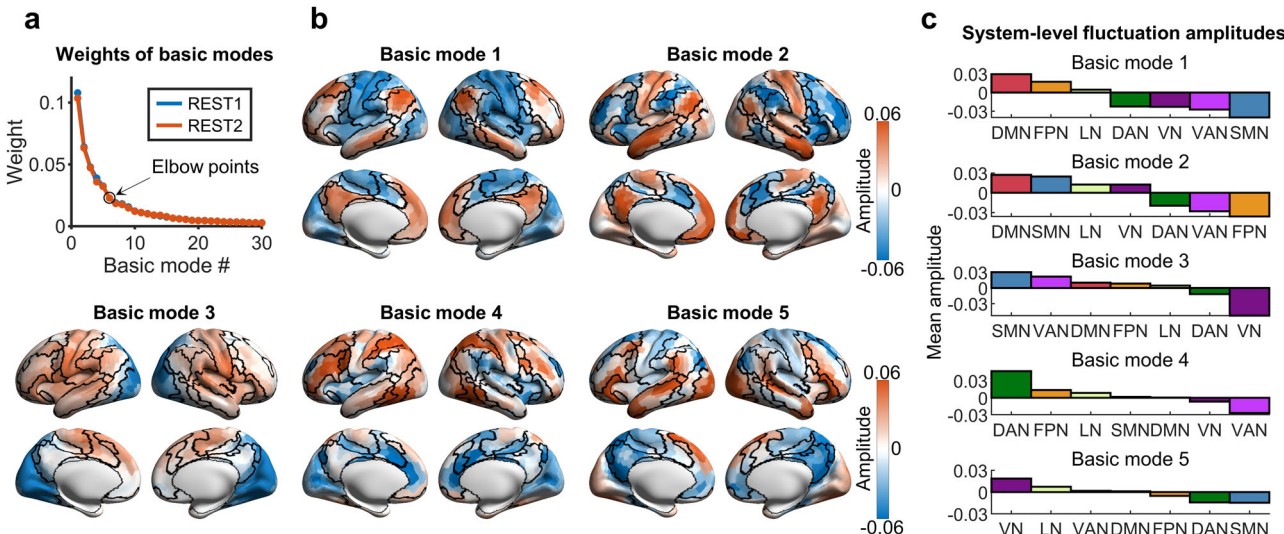

**Fig. 1 Leading basic modes of spontaneous brain activity. a** Weights of basic modes for two runs (i.e., REST1 and REST2). The weights of the first thirty basic modes are displayed. Similar decreasing trends were observed for both runs. The first five basic modes were defined as the leading basic modes according to the criteria modified from a previous study[95]. **b** Spatial patterns of the first five basic modes (i.e., leading basic modes) for REST1. Black curves denote the boundaries of seven functional systems defined in a prior brain template[53]. **c** System-level fluctuation amplitudes for the leading basic modes. Seven functional systems[53] were considered. DMN default-mode network, FPN frontoparietal network, LN limbic network, VAN ventral attention network, DAN dorsal attention network, SMN somatomotor network, VN visual network. The cortical maps were visualized using the toolbox of the BrainNet Viewer[100].

**Relationship between leading basic modes and cognitive functions**. We examined whether spatial patterns of five leading basic modes were related to specific cognitive functions by comparing them with brain activation maps and cognitive components. First, we observed that these leading basic modes corresponded to different profiles of cognitive terms (Fig. 2a) based on the NeuroSynth meta-analytic database[55]. Basic mode 1 was positively associated with the default-mode-related functions and negatively associated with sensorimotor and visual functions. Basic mode 2 was positively associated with the internally oriented and social inference processes and negatively associated with working memory and task-oriented processes. Basic mode 3 was positively associated with sensorimotor, auditory, and language terms and negatively associated with vision-related functions. Basic mode 4 was positively associated with cognitively demanding tasks (i.e., tasks, calculation, and objects) and negatively associated with pain-related terms. Basic mode 5 showed positive associations with the semantic-related functions and negative associations with the default-mode-related functions.

We also examined the spatial similarity between the five leading basic modes and 12 cognitive components that represent fundamental activation components during task performance[56] (Fig. 2b). Statistical significance of these spatial similarities was corrected for spatial autocorrelation (all $ps < 0.05$, 10,000 permutations)[57]. Basic mode 1 was associated with internal mentation, emotion, interoception, and hand and face-related sensorimotor functions. Basic mode 2 was associated with several higher-order cognitive functions, including internal mentation, working memory, inhibition, interoception, and dorsal attention. Basic mode 3 was associated with both externally- and internally-oriented perceptions. Basic mode 4 was involved in working memory, dorsal attention, inhibition, reward, and interoception. Basic mode 5 was associated with visual, auditory, and language functions. Overall, the first three leading basic modes are relevant to the internally oriented, executive-control, and primary cognitive functions, whereas the latter two leading modes are related to more sophisticated and abstract functions.

**Leading basic modes captured individual-specific FC patterns**. Given that the leading basic modes served as the fundamental spatial components for whole-brain activity, we hypothesized that they would make a dominant contribution to the whole-brain FC pattern and capture the individual-specific functional organization. To test this hypothesis, we developed a theoretical model based on the eigen-microstate analysis (Fig. 3a), which decomposed the whole-brain FC matrix into a weighted superposition of the coactivation patterns in the basic modes. This model indicates that each basic mode corresponds to a specific FC pattern (Supplementary Fig. 3). We reconstructed the whole-brain FC matrix based on this model by considering different numbers of basic modes and then compared it with the original FC matrix obtained as Pearson's correlations between nodal time courses. At the population level, the spatial similarity between the reconstructed and original FC matrices slowly increased with the number of basic modes considered and then reached a plateau (Fig. 3a and Supplementary Fig. 4a). Specifically, we observed a high spatial similarity between two FC matrices when including the five leading basic modes ($rs = 0.95$ for both REST1 and REST2, $ps < 0.001$) (Fig. 3b and Supplementary Fig. 4b). Similar results were observed at the individual level. The spatial similarity between the reconstructed and original FC matrices was high for all participants when considering the five leading basic modes (Fig. 3c, mean ± std = 0.94 ± 0.02 for REST1 and 0.93 ± 0.02 for REST2).

We further evaluated whether the leading basic modes captured individual-specific functional organization. First, we examined the reliability of individual reconstructed FC matrices between two runs. We found that the reconstructed FC matrix showed significantly higher values in the intra-subject similarity than inter-subject similarity, regardless of the number of basic modes used (Fig. 3d, all $ps < 0.05$). Then, we performed the individual identification analysis[13] by comparing individual FC matrices between two runs. We observed an identification accuracy of 97% based on the original FC matrix. For reconstructed individual FC matrices, the identification accuracy increased rapidly with increasing number of basic modes and

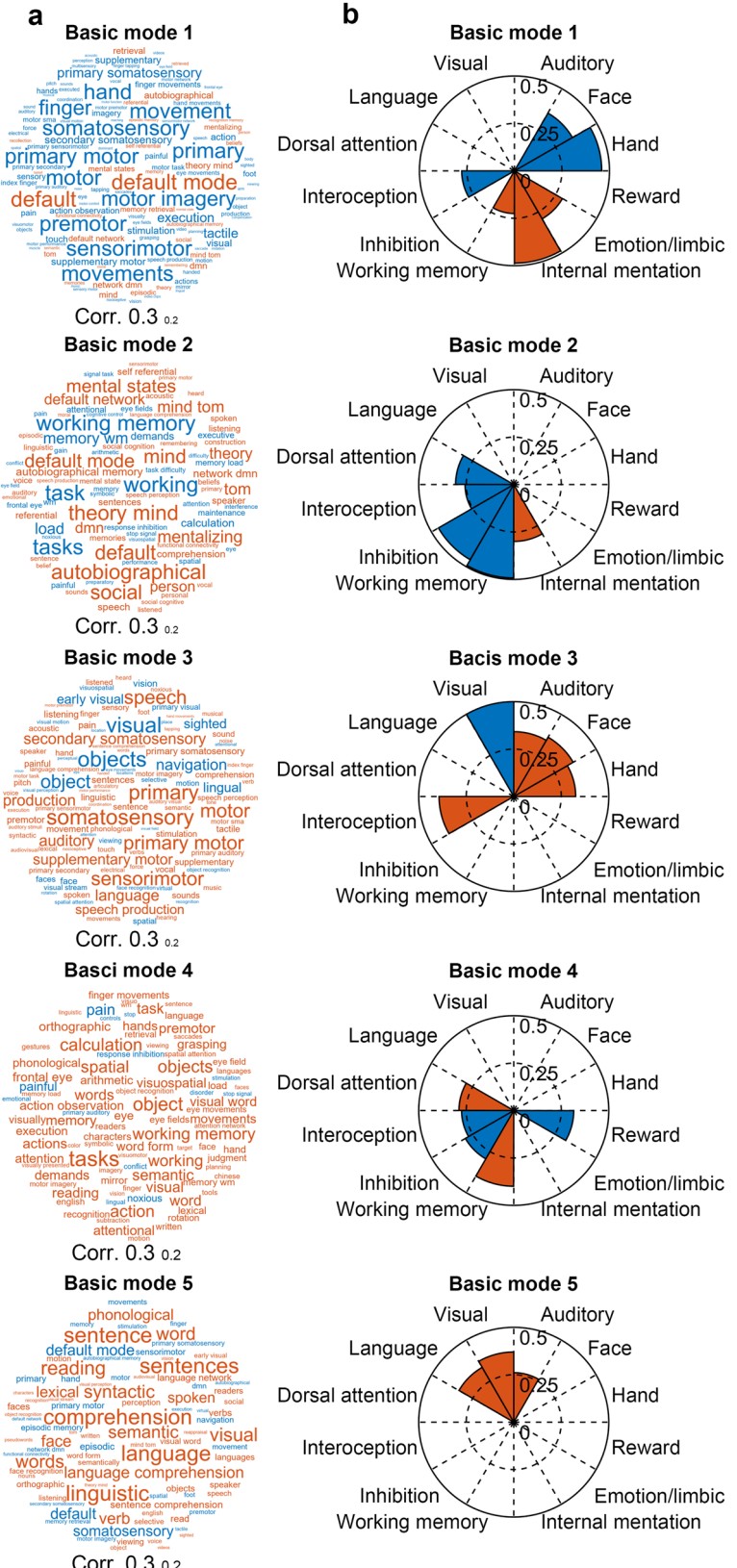

**Fig. 2 Association with cognitive functions. a** Cognitive term associated with each leading basic mode. These terms were obtained based on the NeuroSynth meta-analytic database[55]. Font sizes of cognitive terms denote correlation values between the corresponding cognitive term maps and the leading basic modes. **b** Associations with 12 cognitive components for each leading basic mode. The 12 cognitive components were derived from Yeo et al. [56]. In **a**, **b**, red, and blue colors denote positive and negative correlations, respectively.

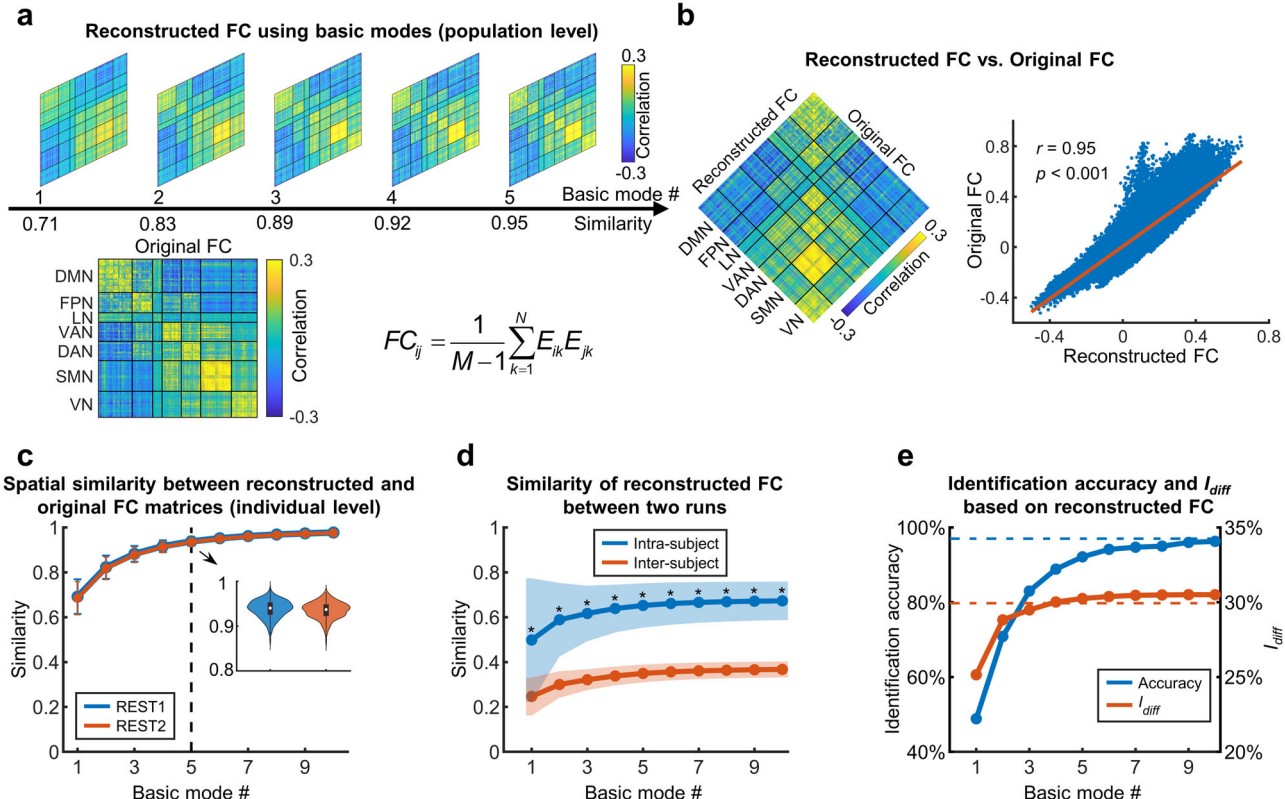

**Fig. 3 Reconstructing functional connectivity based on the basic modes. a** Spatial similarity between the reconstructed and original FC matrices at the population level (REST1). The original FC matrix was estimated based on the concatenated time courses of all participants. The reconstructed FC matrix was generated separately by using different numbers of basic modes based on the theoretical model. $M$ denotes the number of time points, $N$ denotes the number of all possible basic modes, and $E_{ik}$ is the $i$th element of the $k$th basic mode. **b** Reconstructing the FC matrix with the first five basic modes (REST1). Left, spatial patterns of the reconstructed and original FC matrices; right, spatial similarity between these two matrices. **c** Spatial similarity between the reconstructed and original FC matrices at the individual level for both runs. The reconstructed FC matrix was generated by using different numbers of basic modes. Mean spatial similarity across participants and the corresponding standard deviation are displayed ($n = 700$ participants). The distributions of individual spatial similarity obtained from the five leading basic modes are shown in the subplot in the form of the violin plots and the box plots. **d** Intra- and inter-subject similarity of the reconstructed FC matrices between two runs. Mean similarity across participants and the corresponding standard deviation are displayed ($n = 700$ participants). *, significant differences (paired $t$-tests, $ps < 0.05$). **e** Individual identification accuracy and differential identifiability $I_{\text{diff}}$ based on the reconstructed FC matrix. Individual FC matrices were reconstructed with different numbers of basic modes. The dashed lines denote the identification accuracy and $I_{\text{diff}}$ based on the original FC matrix ($n = 700$ participants). FC functional connectivity, DMN default-mode network, FPN frontoparietal network, LN limbic network, VAN ventral attention network, DAN dorsal attention network, SMN somatomotor network, VN visual network.

reached 92% when the five leading basic modes were included (Fig. 3e). Similarly, the differential identifiability $I_{\text{diff}}$, which quantifies the difference between the mean intra-subject similarity and the mean inter-subject similarity[58], also increased rapidly with increasing numbers of basic modes. Its value reached 30.3% when all the five leading basic modes were included, which exceeded $I_{\text{diff}}$ of the original FC matrix (Fig. 3e). These results suggest that these leading basic modes make the dominant contribution to the individualized functional organization.

**Influence of sleep deprivation on the leading basic modes**. To assess whether the leading basic modes are affected by mental states, we applied the eigen-microstate analysis to the sleep deprivation dataset[50]. In this dataset, 19 participants underwent R-fMRI scanning during rested wakefulness and after sleep deprivation. Similar to the HCP dataset, the weights of the basic modes at rested wakefulness decreased rapidly with increasing ranking and reached the elbow point at the 7th basic mode (Fig. 4a), indicating the presence of a small set (i.e., six) of leading basic modes (Supplementary Fig. 5). These six leading basic modes showed a spatial correspondence with the first six basic

modes of the HCP dataset, except for an inversion between basic mode 2 and basic mode 3 (Fig. 4a, all $rs > 0.78$).

Next, we evaluated the influences of sleep deprivation by comparing the leading basic modes between two mental states (i.e., at rested wakefulness and post-sleep deprivation). After the post-sleep deprivation, we identified seven leading basic modes (Fig. 4b and Supplementary Fig. 6). The spatial patterns of these seven basic modes showed a spatial correspondence between two states, except for an inversion between basic mode 4 and basic mode 5 (Fig. 4b). Notably, all these modes showed relatively low similarity values between the two states, except for the first three modes and the 6th mode.

We further examined the difference in spatial patterns for the four basic modes (i.e., 1st, 2nd, 3rd, and 6th) that maintained spatial correspondences between the two states. Significant changes in amplitude were observed in basic mode 1 (Fig. 4c, $p < 0.05$, false discovery rate (FDR) corrected, 10,000 permutations). Significant increases were primarily located in regions of the frontoparietal, ventral, and dorsal attention and lateral visual networks, while significant decreases were mainly located in regions of the default-mode network. Interestingly, for most (86%) of these regions, the directions of the amplitude changes

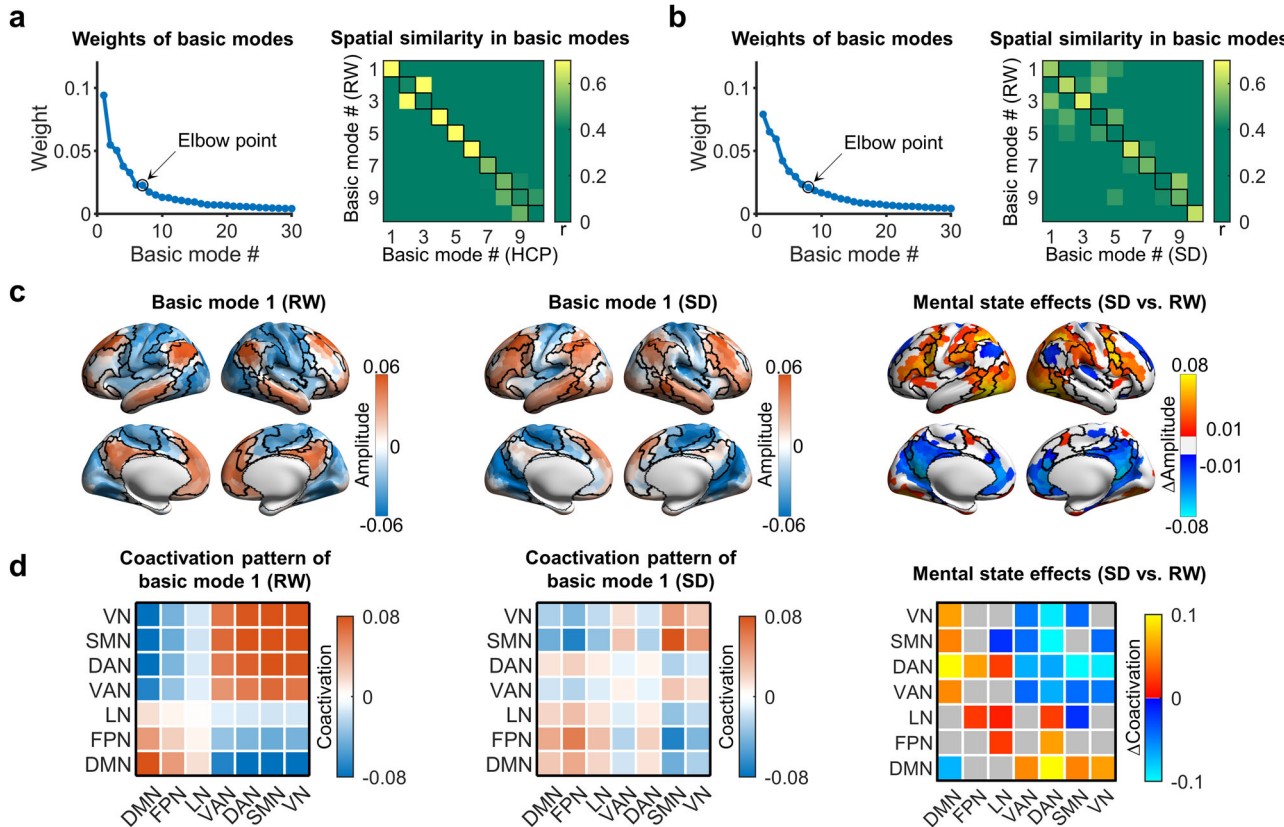

**Fig. 4 Influence of sleep deprivation on the basic modes in the sleep deprivation dataset. a** Weights of basic modes for R-fMRI data at rested wakefulness and spatial similarity of basic modes with REST1 in the HCP dataset. **b** Weights of basic modes for R-fMRI data after sleep deprivation and spatial similarity of basic modes with R-fMRI data at rested wakefulness. In **a**, **b**, the spatial similarity was estimated for every pair of basic modes between two conditions to examine the spatial correspondence. **c** Spatial patterns of basic mode 1 at rested wakefulness and after sleep deprivation and their differences. Regions showing significant changes were detected with a permutation test ($p < 0.05$, FDR corrected, $n = 19$ participants). **d** System-level coactivation pattern of basic mode 1 at rested wakefulness and after sleep deprivation and between-state differences. Significant changes were detected at the system level with a permutation test ($p < 0.05$, FDR corrected, $n = 19$ participants). RW rested wakefulness, SD after sleep deprivation, HCP Human Connectome Project.

were opposite to the signs of the original amplitudes (Supplementary Fig. 7), suggesting that the spatial inhomogeneity of brain activity was reduced after sleep deprivation. At the connectivity level, we also observed significant changes in the coactivation pattern of basic mode 1. The significant increase was mainly located between the default-mode network and the primary and attention networks, while the significant decrease was mainly located between the attention and the primary networks as well as within the default-mode network (Fig. 4d, $p < 0.05$, FDR corrected, 10,000 permutations), further supporting the reduction of cross-system inhomogeneity. Unlike basic mode 1, no significant changes were observed for basic modes 2, 3, and 6 after sleep deprivation (all $ps > 0.05$, 10,000 permutations).

**Validation results**. We further assessed the reliability of the presence and spatial patterns of the leading basic modes (Supplementary Figs. 8–13). Five additional analysis strategies were considered, including (i) using different numbers of participants; (ii) using stricter head motion exclusion criteria; (iii) performing nuisance regression without global signal regression; (iv) defining brain nodes based on two functional parcellations with different spatial resolutions; and (v) using another independent dataset of 197 participants, i.e., the Beijing Zang dataset[51]. The number of the leading basic modes reached five without variation when the participant number considered was equal to or greater than 250 (Supplementary Fig. 8). Meanwhile, spatial patterns of these

leading basic modes were highly similar to those obtained from the whole population (i.e., 700 participants) with one-to-one correspondence (all mean $rs > 0.96$, Supplementary Fig. 8). The presence of five leading basic modes was replicated with high spatial similarity in most of the other cases (all $rs > 0.85$, Supplementary Figs. 9, 11, and 12), except for the case of without global signal regression (Supplementary Fig. 10). Notably, the total weight explained by the leading basic modes increased with the decreasing spatial resolution (Supplementary Fig. 13), with weights of 37% and 44% for the 400-node and 200-node parcellations, respectively. For the strategy without global signal regression, the number of leading basic modes was reduced to three (Supplementary Fig. 10). The reduced number might be biased by the presence of an additional basic mode, which ranked ahead of the five typical basic modes. This additional basic mode showed all positive amplitudes across the brain and accounted for a large portion of activity variance (i.e., 23%). All these results suggest that the five leading basic modes were robust and reproducible.

## Discussion

Using the recently proposed eigen-microstate analysis from statistical physics theory, this study reveals the presence of a few leading basic modes that dominate the temporal fluctuations of spontaneous activity. The leading basic modes exhibited distinct and cognitive function-specific spatial patterns, suggesting the

coexistence of multiplexed coactivation relationships between regions. Furthermore, these leading basic modes dominantly contributed to individual whole-brain FC patterns and were modulated by sleep deprivation. Taken together, our findings highlight a small set of leading basic modes that dominate spontaneous activity and demonstrate their functional significance, opening an avenue to explore multiplexed interregional relationships in the healthy and diseased brain.

We note that the eigen-microstate analysis shares similarities with serval existing methods that are used to decode the spatiotemporal organization of spontaneous brain activity. For example, several component analysis methods, such as the principal component analysis (PCA)[59], independent component analysis (ICA)[60], and temporal functional mode analysis[61], have been applied to fMRI time series to identify dominant interregional interaction patterns. The specified spatial components are often regarded as functional networks[59,62,63], while their spatial patterns reflect the relative weights of brain regions. Some other relevant approaches have detected brain states or modes by considering dynamic FC patterns[30,45,64–66]. Each connectivity state differs from the other in terms of the overall connectivity pattern[64] or dominant connectivity modes quantified by leading eigenvectors[65,67]. Different from these previous approaches, the eigen-microstate analysis focuses primarily on spatial patterns of instantaneous activity per se and further establishes a bridge between instantaneous brain activity and FC patterns. Moreover, the eigen-microstate analysis assumes that multiple basic modes may coexist over time rather than a dominant brain state at each time, which allows for capturing delicate changes in brain activity over time.

In this study, we identified a small set of fundamental brain activity patterns (i.e., leading basic modes) that dominate rich repertoires of spontaneous activity, regardless of datasets or mental states. These results suggest a reliable low-dimensional representation of seemingly complicated spontaneous activity. A low-dimensional representation of spontaneous activity has also been reported for rat cortical activity[68] and human brain activity across multiple task states[69]. We also found that each leading basic mode was associated with different cognitive functions. The first three leading basic modes were associated with fundamental functions that are necessary for daily life, such as sensorimotor, visual, internally-oriented, and executive-control functions. The next two leading basic modes were associated with more sophisticated and abstract cognitive functions, such as calculation, reward, and language-related items. These findings are consistent with previous hypotheses aimed at interpreting the biological significance of time-resolved activity patterns[24,41]. These hypotheses argue that spontaneous brain activity may transit between a number of general priors, which are low-dimensional representations of typical behavioral states in past experience[41] and are selected at different moments for an efficient and flexible cognitive response[24,41]. In this sense, the leading basic modes observed here might serve as the general priors, and their weights may reflect the frequencies of the corresponding behaviors in past experiences.

Within the framework of the eigen-microstate analysis, distinct spatial patterns of these leading basic modes indicate the coexistence of distinct coactivation (i.e., coordination) patterns between regions. Interestingly, the first and second leading basic modes showed distinct relationships between the default-mode network, two cognitive control networks (i.e., frontoparietal and attention networks), and the primary networks. The first leading basic mode shows an anti-correlation primarily between the default-mode network and the primary and attention networks, which is highly similar to the previously reported spatial pattern of the principal gradient of the whole-brain FC pattern[54]. This

finding suggests that the separation of brain activity follows the hierarchical organization of information processing[70]. The second leading basic mode shows a separation of activity between the default-mode and cognitive control networks, which would explain the commonly observed alternative activities or anti-correlations between the default-mode and task-positive regions over time[71,72]. Compared to the previous assumption of one state per time point[31,64], the coexistence of multiple leading basic modes here suggests a parallel information processing between regions at each time point, offering fresh insights into time-varying connectivity patterns[26].

The presence of the leading basic modes in intrinsic activity might be shaped by anatomical substrates of the brain, given the tight structure-function coupling of the brain[73,74]. Previous studies have demonstrated that spatial arrangements of cortical microstructures show a dominant gradient spanning between sensorimotor-to-transmodal areas[75]. For example, the myelination map of the brain[76] shows spatial similarities with the first three leading basic modes observed here, indicating a potential link between the macroscale brain activity and the local microstructure. However, how these leading basic modes emerge from the anatomical properties, such as myelination, cortical thickness, and white-matter connectivity, requires further investigation.

Recent studies have reported that resting-state FC is driven by instantaneous brain activity at several critical time points[28,44], implicitly ignoring interregional coordination at other time points. Here, we used the leading basic modes, which were identified from full repertoires of spontaneous activity, to bridge the gap between instantaneous activity and the FC pattern. A theoretical model was developed to recover the FC pattern as a weighted superposition of the coactivation patterns of these leading basic modes. This model suggests that each basic mode corresponds to a specific FC pattern and that multiplexed relationships (i.e., parallel communication) exist simultaneously between regions. Our idea is consistent with a recent study showing that the individual FC pattern can be attributed to the contribution of multiple factors (e.g., group, individual, and task)[77], but it further clarifies the origins of FC patterns by providing detailed patterns of the candidate components. Interestingly, the five leading basic modes, which account for 29% of the total weight, can be used to reconstruct the original FC pattern with a high spatial similarity ($r = 0.95$). This seemingly contradictory finding suggests that these leading basic modes may capture the intrinsic coordination behavior of spontaneous activity, while the remaining basic modes may be vulnerable to unconstrained cognitive activity, head motion, or other perturbations and thus make small contributions to interregional coordination.

The leading basic modes identified here show intriguing spatial similarities with dominant connectivity patterns or modes reported in previous studies[54,65,67]. Specifically, the first three leading basic modes exhibit consistent patterns with the first three gradients of the cortical FC pattern previously identified[54]. Similarly, the Leading Eigenvector Dynamics Analysis (LEiDA) approach identified typical connectivity modes in healthy older adults[65], such as one mode with global coherence across the brain and one mode with high coherence within the default-mode network. These two modes align with the leading basic modes in our study when the global signal is retained. Moreover, the dynamic mode decomposition identified typical FC modes with different temporal features (e.g., damping time and oscillatory periods)[67]. The first dynamic mode, characterized by an anti-correlation between the default mode and task-positive networks, partially overlaps with the second leading basic mode in our study. Similar patterns between different types of maps further support the notion that leading basic modes play a dominant role

in shaping the FC pattern. The discrepancies observed in other modes across studies may be attributed to differences in brain coverage, node definitions, and specific populations and features of interest examined in each study. These methodological variations should be considered when interpreting and comparing results across studies.

Moreover, we found that the reconstructed FC patterns based on the five leading basic modes captured individual uniqueness in functional organization, providing additional clues for understanding individual differences in functional organization. Notably, the FC pattern reconstructed by only the first leading basic mode was highly similar to the original pattern but showed a moderate performance in individual identification and a relatively low differential identifiability value. These results suggest that the first leading basic mode may serve as a backbone or a group factor of brain activity shared across individuals, as suggested by Gratton et al.[77]. Identification accuracy and differential identifiability increased rapidly when subsequent basic modes were included, indicating that more individual-specific information is captured by subtly modulating the backbone of the FC pattern.

Sleep deprivation was used as a modulating factor to assess the influence of the mental state on the leading basic modes. A small number of leading basic modes were also identified after sleep deprivation, indicating the reliability of low-dimensional representations of spontaneous brain activity regardless of mental state. The spatial patterns of the first three and the sixth leading basic modes remained similar after sleep deprivation, whereas the other leading basic modes remarkably changed. This finding is consistent with a recent study showing that the first three FC gradients remain largely unchanged after sleep deprivation[78]. Our results suggest that the activity representations relevant to fundamental cognitive functions might be reliable across mental states, while those related to more sophisticated and abstract functions may be more vulnerable. The different sensitivities of the leading basic modes of mental state may be valuable for future studies investigating functional organization across states.

In addition, a regional comparison revealed that the spatial heterogeneity of the first leading basic mode was reduced after sleep deprivation, manifested as a weakened separation of activity between the default-mode and task-positive areas (e.g., attention and somatomotor networks). This observation is confirmed by comparing the coactivation patterns corresponding to the first leading basic mode between two states. These results are consistent with previous findings that sleep deprivation is associated with the failure of the default-mode network to remain functionally distinct from its anti-correlated networks, i.e., task-positive networks[79–81]. This impairment in the decoupling between the default-mode network and task-positive networks may be further related to participants' cognitive vulnerability to sleep deprivation[80], but more evidence is needed to support this idea. Notably, the 4th and 5th leading basic modes showed remarkable reconfiguration after sleep deprivation. Given that these two modes are associated with sophisticated and abstract cognitive functions, the reconfiguration of these modes indicates the sensitivity of these functions to sleep deprivation.

Several issues should be further considered. First, we identified an additional basic mode with all positive amplitudes when identifying leading basic modes without global signal regression in the data preprocessing. This suggests that preserving the global signal may enhance coactivations between regions, providing a clear explanation for the usually observed rightward shift in the distribution of FC strength compared to the case with global signal regression[82]. Second, we explored the potential cognitive significance of leading basic modes through association analysis. In future studies, it is suggested to investigate the changes in the leading basic modes across task states to provide more direct

clues. Third, additional leading basic modes were identified in the sleep-deprivation dataset, which may be influenced by several factors, such as the small sample size or mental states. A larger dataset is needed to further confirm the potential effects of sleep deprivation. Fourth, the biological origins of the leading basic modes remain unclear. Recent computational models of large-scale brain circuits have found that both interregional white matter connections and local circuit properties can shape resting-state FC and its itinerant dynamics[83,84]. In the future, the computational modeling approach can be used to explore the relationship between the leading basic modes and the underlying structural network and local morphological properties of the human brain. Finally, the eigen-microstate analysis used here is essentially a linear decomposition, which implicitly assumes that the rich repertoire of brain activity can be embedded in a low-dimensional linear subspace spanned by the leading basic modes. However, the biological plausibility of the low-dimensional nonlinear representation of spontaneous brain activity deserves further investigation.

## Methods

**Participants and study design**. We employed three datasets of R-fMRI data from healthy young adults. The first dataset consisted of multiband R-fMRI data from 970 participants from the publicly available S900 data release of the Human Connectome Project (HCP)[49]. The scanning protocol of the HCP dataset was approved by the Institutional Review Board at Washington University. These subjects underwent repeated R-fMRI runs in two sessions. The second dataset, named the sleep deprivation dataset, included repeated R-fMRI data from 20 participants scanned separately during rested wakefulness and after sleep deprivation[50]. The research was approved by the Institutional Review Board of the Institute of Biophysics (Chinese Academy of Sciences). The third dataset, named the Beijing Zang dataset, included R-fMRI data from 198 participants selected from the 1000 Functional Connectomes Project[51]. The research was approved by the Institutional Review Board of the State Key Laboratory of Cognitive Neuroscience and Learning at Beijing Normal University. Informed consent was obtained from all participants of the three datasets. All ethical regulations relevant to human research participants were followed. The first two datasets were used for the main analysis, and the third dataset was used for the replication analysis.

**Data acquisition**. In the HCP dataset, all participants underwent multimodal MRI scanning with a customized 32-channel SIE-MENS 3 T Connectome Skyra scanner at Washington University, USA. Four multiband R-fMRI runs were acquired in two sessions for each participant. Briefly, each session consisted of two runs that were separate phases encoded in the left-to-right and right-to-left directions. The R-fMRI scans were obtained using a multiband gradient-echo-planar imaging sequence (repetition time [TR] = 720 ms and 1200 volumes per run, i.e., 14.4 min), with participants' eyes fixated on a bright projected crosshair. Here, we used only the left-to-right-encoded scans to reduce the potential influence of the phase-encoding directions[15,85]. In the original S900 data release, 837 participants completed the left-to-right-encoded R-fMRI scans in both sessions, denoted as REST1 and REST2 separately. Of these, 137 participants were excluded due to missing time points ($N = 10$), excessive head motion ($N = 105$) (see "Data Preprocessing"), and arachnoid cysts ($N = 22$). Data from the remaining 700 participants (aged 21–35 years, M/F: 304/396) were used for the main analyses.

In the sleep deprivation dataset, 20 participants underwent repeated R-fMRI scans separately during rested wakefulness and

after sleep deprivation (for design details, see Zhou et al.[50]). All the participants were right-handed and had no history of neuropsychiatric disorders. The MRI data were acquired using a 64-channel 3 T Siemens Prisma scanner at the Beijing MRI Center for Brain Research of the Chinese Academy of Sciences. R-fMRI data were acquired using a T2*-weighted gradient-echo-planar imaging sequence (TR = 1000 ms and 480 volumes per run), with participants' eyes fixated on a crosshair. Structural images were acquired using a 3D T1-weighted, magnetization-prepared rapid acquisition gradient-echo sequence. One participant was excluded due to excessive head motion (see "Data Preprocessing"). Data from the remaining 19 participants (aged 18–26 years, M/F: 7/12) were used for the main analysis.

For the Beijing Zang dataset, 198 participants underwent MRI scanning using a 12-channel Siemens Trio Tim 3.0 T scanner in the Imaging Center for Brain Research, Beijing Normal University. R-fMRI data were acquired with participants' eyes closed (TR = 2000 ms and 235 volumes). One participant was excluded due to differences in scanning orientation, leaving 197 participants (aged 18–26 years, M/F: 75/122) used for the cross-validation analysis.

**Data preprocessing**. For the HCP dataset, we employed the minimally preprocessed R-fMRI data[86], followed by ICA-Fix denoising[87]. Four additional steps were performed using the GRETNA package[88], including the removal of the first 10-second volumes (i.e., 15 volumes), linear detrending, nuisance regression, and temporal filtering (0.01–0.08 Hz). During the nuisance regression, white matter, cerebrospinal fluid, and global signals were included as regressors to further remove the influence of head motion and physiological noise[89].

The sleep-deprivation dataset and the Beijing Zang dataset were preprocessed with the same pipeline using the GRETNA package[88]. Specifically, the preprocessing included the removal of the first 10-s volumes, realignment, spatial normalization to the Montreal Neurological Institute (MNI) space with the T1-unified segmentation algorithm[90], linear detrending, nuisance regression, and temporal filtering (0.01–0.08 Hz). During the nuisance regression, we included Friston's 24 head-motion parameters[91], white matter, cerebrospinal fluid, and global signals as regressors to reduce the influence of head motion and physiological noise[89].

For these three datasets, we excluded participants with excessive head motion in any scan, including a translation/rotation greater than 3 mm or 3° and a mean framewise displacement (FD) over time[92] greater than 0.5 mm. After applying these criteria, 105 participants were excluded from the HCP dataset, and 1 participant was excluded from the sleep-deprivation data.

**Eigen-microstate analysis of spontaneous brain activity**. We applied a recently proposed eigen-microstate analysis[47,48] from the statistical physics theory to the HCP dataset to identify basic modes underlying the rich repertoire of brain activity. The eigen-microstate analysis is an approach specifically designed to extract basic activity modes (i.e., eigen-microstates) of the temporal evolution of a system. This approach has been applied to several complex systems (e.g., the Earth system and stock markets) and reveals meaningful activity modes that show well-defined spatial patterns[47]. In this study, the eigen-microstate analysis was applied to the time courses of each R-fMRI run (i.e., REST1 and REST2) separately. The brain microstates were defined at the nodal level to reduce computational burden.

*Definition of the ensemble matrix and microstates.* First, we defined 1000 cortical nodes based on a prior functional

parcellation[52], which would enhance functional homogeneity within each nodal region. Then, we extracted the time courses of these nodes for each participant. The time course for each node was further transformed into z-score values with zero mean and unit variance over time. Finally, the normalized nodal time courses were concatenated across participants, resulting in an $N \times M$ time course matrix $A$, where $N$ denotes the number of nodes (i.e., 1000 here) and $M$ denotes the number of time points in the concatenated time courses (i.e., $1185 \times 700$). Matrix $A$ was considered as an ensemble matrix representing a rich repertoire of brain activity, and each column, $A_t$, represents a microstate of brain activity at a specific time point from a statistical physics perspective.

*Theoretical derivation of eigen-microstates.* The eigen-microstate analysis assumes that microstates may exhibit commonalities over time, with instantaneous brain activity arising from the combination of multiple basic modes. It aims to identify the basic modes (i.e., eigen-microstates) that represent fundamental building blocks of rich microstates and are independent of each other[47,48]. Specifically, we first calculated the covariance matrix between microstates as $C = A^T A$[47], which reflects the spatial similarity between all microstates. Next, we computed the eigenvectors of the matrix $C$,

$$C\boldsymbol{v}_i = \lambda_i \boldsymbol{v}_i, \tag{1}$$

wherein $\boldsymbol{v}_i$ is the $i$th eigenvector of the matrix $C$, and $\lambda_i$ is the corresponding eigenvalue. The representativeness of spatial patterns of the microstates at different time points is captured by the eigenvectors[93], which take into account the entire spatial similarity pattern between all microstates. Finally, we obtained each eigen-microstate as the weighted sum of the origin microstates:[47,48]

$$\boldsymbol{E}_i = A\boldsymbol{v}_i = \sum_{t=1}^{M} \boldsymbol{A}_t v_{ti}, \tag{2}$$

wherein $v_{ti}$ denotes the $t$th element of eigenvector $v_i$ and quantifies the contribution of $t$th microstate to the $i$th eigenvector. In other words, the basic modes (i.e., eigen-microstates) were identified by incorporating brain activity patterns over time and thus can be intuitively interpreted as typical activity patterns. The weight factor of a given eigen-microstate $\boldsymbol{E}_i$ is defined as $w_i = \boldsymbol{E}_i^T \boldsymbol{E}_i = \lambda_i$.

*Calculation of eigen-microstates.* In practical analysis, it is difficult to derive the eigenvectors of the covariance matrix $C$ one by one due to the large number of time points $M$. Therefore, we obtained the eigen-microstates with the aid of the singular value decomposition (SVD)[47]. Based on the SVD, the ensemble matrix $A_{N \times M}$ was factorized as the product of three matrices:

$$A_{N \times M} = U_{N \times N} \Sigma_{N \times M} V_{M \times M}^T, \tag{3}$$

wherein $U = [\boldsymbol{u}_1, \boldsymbol{u}_2, …, \boldsymbol{u}_N]$ and $V = [\boldsymbol{v}_1, \boldsymbol{v}_2, …, \boldsymbol{v}_M]$ contain eigenvectors for two matrices of $AA^T$ and $A^T A$, respectively, and $\Sigma_{N \times M}$ is a diagonal matrix of singular values ($\sigma_i$). By substituting this decomposition into Eq. (2), the definition of the eigen-microstate can be reformulated as:

$$\boldsymbol{E}_i = A\boldsymbol{v}_i = \sigma_i \boldsymbol{u}_i, \tag{4}$$

wherein $\boldsymbol{u}_i$ is the eigenvector of the matrix $AA^T$. The weight of each eigen-microstate can be obtained as $w_i = \boldsymbol{E}_i^T \boldsymbol{E}_i = \sigma_i^2$. The matrix $A$ was normalized by dividing the root-sum-square of all its elements (i.e., a dataset-dependent constant $S$) before SVD to ensure $\Sigma \sigma_i^2 = 1$.

Of note, the estimation for eigen-microstates in Eq. (4) is similar to the PCA[94]. However, the eigen-microstate analysis and PCA are used in different contexts. The former is specific to the

temporal evolution of a system and is used to identify the typical spatial patterns of microstates by analyzing the spatial relationship between time points (see Eq. (2)). The latter, however, is a technique commonly used to reduce the dimensionality of a dataset by finding the principal components that capture the main variance in the multivariate data. In this sense, PCA focuses on the relationship between data variables (here, brain regions) rather than the relationship between time points. Moreover, compared to PCA, which only captures the relative weights of variables, eigen-microstate analysis can provide an intuitive biological interpretation of the eigen-microstates as typical activity states (see Eq. (2)).

To determine whether the brain is dominated by a small number of basic modes (i.e., low-dimensional representations), we identified leading basic modes, whose weights should be[95]: (i) substantially greater than the weight of the subsequent basic mode, known as Cattell's scree test[96]; (ii) greater than the average weight across all $N$ possible basic modes, i.e., $1/N$; (iii) statistically significant according to a permutation test. In each permutation instance, the labels of nodal regions at each time point were randomly shuffled to disrupt the spatial organization. The statistical significance level of each leading basic mode was determined by comparing its original weight (i.e., $\sigma_i{}^2$) to the null distribution of the corresponding weights obtained from the 10,000 permutation instances. In our analysis, Cattell's scree test was performed by identifying the elbow point on the weight curve according to the Kneedle algorithm[97] (https://github.com/arvkevi/kneed).

We further investigated the spatial patterns of the leading basic modes based on a prior functional system definition[53]. Seven systems were considered, including the default-mode network, frontoparietal network, limbic network, ventral attention network, dorsal attention network, somatomotor network, and visual network. For each leading basic mode, we estimated the mean fluctuation amplitude for each system by averaging the nodal values within this system.

We also performed the eigen-microstate analysis for each participant to investigate the presence of the leading basic modes at the individual level. In this condition, matrix $A$ in the above analysis was replaced as the normalized time course within each participant for each R-fMRI run.

**Cognitive function associations of the leading basic modes**. We investigated the potential functional roles of the leading basic modes from two perspectives. First, we examined the association between these leading basic modes and cognitive functions based on the NeuroSynth meta-analytic database (www.neurosynth.org)[55]. For each leading basic mode, we calculated its spatial similarity with all available meta-analytic activation maps using Pearson's correlation across voxels. The associated cognitive terms are illustrated using word cloud plots.

Second, we compared each of the leading basic modes with 12 cognitive components[56]. Each cognitive component represents a basic activation probability map that is involved in various cognitive tasks[56]. For each cognitive component, we estimated the corresponding node-level version by averaging the activation probabilities of all voxels within each node. We then calculated the spatial similarity between each of the leading basic modes and the 12 cognitive components by using Pearson's correlation across nodes. To correct for the potential influence of spatial autocorrelation, the statistical significance of each spatial similarity was tested using a permutation test ($n = 10,000$). The significance level was determined by comparing the original similarity to the null distribution of the corresponding similarity obtained from 10,000 permutation instances. For each

permutation instance, we generated a surrogate basic mode map that preserved the spatial autocorrelation of the original basic mode[57].

**Relationship between leading basic modes and FC**. Since the leading basic modes dominated the spontaneous fluctuations of brain activity, we further investigated how they contribute to the FC between regions.

The original FC between two nodal regions is defined as Pearson's correlation between their time courses[4]. As each regional time course ($A_{it}$, $t = 1, \dots M$) was normalized over time (i.e., mean $= 0$ and SD $= 1$), the FC between nodes $i$ and $j$ can be estimated as:[13,43]

$$\text{FC}_{ij} = \frac{1}{M-1} \sum_{t=1}^{M} A_{it} A_{jt}, \qquad (5)$$

where $M$ denotes the number of time points in the time course.

By substituting Eq. (3) and Eq. (4) into Eq. (5) and considering the time independence between basic modes, we found $FC_{ij}$ between nodes $i$ and $j$ can be rewritten as:

$$\text{FC}_{ij} = \frac{1}{M-1} \sum_{k=1}^{N} \sigma_k^2 u_{ik} u_{jk} = \frac{1}{M-1} \sum_{k=1}^{N} E_{ik} E_{jk}, \qquad (6)$$

where $N$ is the number of all possible basic modes, and $E_{ik}$ is the $i$th element of the $k$th basic mode. Thus, the FC between the two nodes can be attributed to the joint contribution of their coactivation patterns in each basic mode. Notably, the $FC_{ij}$ estimated from Eq. (6) should be further multiplied by a constant $S^2$ to correct for the normalization effect of matrix $A$ prior to the SVD analysis.

To validate the effectiveness of the above theoretical model (i.e., Eq. (6)), we reconstructed the FC matrix according to Eq. (6) by gradually increasing the number of basic modes of interest. We then compared the spatial similarity between the reconstructed and original FC matrices. The spatial similarity was quantified with Pearson's correlation across the lower triangular elements in the matrices. Specifically, we reconstructed the FC matrix at both the population and individual levels. At the population level, the leading basic modes were obtained from the concatenated normalized time course across all participants. At the individual level, the leading basic modes were identified from the time courses of each participant. We then calculated the similarity between the reconstructed and the original FC matrices for each individual.

We further explored whether the basic modes, especially these leading basic modes, could capture the individual functional organization. First, we estimated the reliability of the reconstructed FC matrix between two runs at the individual level. Given a participant of interest, we evaluated the intra-subject similarity of the reconstructed FC matrices between two runs. We also estimated the inter-subject similarity of this subject as the averaged spatial similarity of this participant in the first run (i.e., REST1) with all the other participants in the second run (i.e., REST2). Next, we examined the individual uniqueness in the reconstructed FC matrices by performing an FC-based individual identification analysis between two runs (i.e., REST1 and REST2)[13]. For each participant, we compared the reconstructed FC matrix of this participant in REST1 with those of all the participants in REST2. If the participant with the highest similarity in REST2 was the same participant given in REST1, the identification was correct; otherwise, it was incorrect. Identification accuracy was defined as the proportion of participants that were correctly identified. A higher accuracy indicates a higher capability of the FC matrix in distinguishing individuals. We also evaluated the differential identifiability[58] to quantify the individual uniqueness of the reconstructed FC matrices. The differential identifiability, $I_{\text{diff}}$, was defined as

$I_{\text{diff}} = (I_{\text{self}} - I_{others}) \times 100$, wherein $I_{\text{self}}$ and $I_{others}$ denoted the mean intra-subject similarity and mean inter-subject similarity between two runs, respectively. A larger $I_{\text{diff}}$ value indicates more individual-specific information is captured. For comparison, individual identification analysis was also performed based on the original FC matrix.

**Influence of sleep deprivation on the leading basic modes**. To assess whether the leading basic modes are affected by mental states, we applied the eigen-microstate analysis to the sleep deprivation dataset[50]. We identified the leading basic modes (see "Eigenmicrostate analysis") separately from the R-fMRI data obtained in the two states represented in the dataset (i.e., rested wakefulness vs. post-sleep deprivation). First, we examined the spatial correspondence of the leading basic modes determined at rested wakefulness with those obtained from REST1 of the HCP dataset to investigate the reproducibility of the leading basic modes. Next, we compared the basic modes obtained at the different states (i.e., rested wakefulness vs. post-sleep deprivation) to examine the potential influence of sleep deprivation.

For each leading basic mode that maintained spatial correspondence between two states, we tested differences in regional fluctuation amplitudes between the two states by using a permutation test ($n = 10,000$). In each permutation instance, the state labels of the R-fMRI data were shuffled for each participant. Multiple comparisons across nodal regions were corrected using the FDR approach (corrected $p < 0.05$)[98]. Given that the basic mode showed significant changes, we further investigated how interregional coactivation patterns differed between the two states. Briefly, we obtained the system-level coactivation pattern for the rested wakefulness and post-sleep deprivation separately. The within-system and between-system coactivation values were obtained by averaging the interregional coactivation values within the same system and between different systems, respectively. Significance levels of differences in the coactivation pattern were also estimated using a permutation test ($n = 10,000$) and corrected for multiple comparisons (FDR corrected $p < 0.05$).

**Validation analysis**. The reliability of the leading basic modes was validated by considering five analysis strategies that may affect the identification of the leading basic modes. In each case, the leading basic modes were re-estimated and compared with those obtained in the main analyses (i.e., REST1 in HCP). (i) Number of participants. We re-identified the leading basic modes by including R-fMRI data from different numbers of participants from the HCP dataset, with the number of participants varying from 50 to 650 with a step of 50. Given a participant number, we randomly sampled participants from all the 700 participants, and this process was repeated 100 times. For each sampling instance, we counted the number of the leading basic modes and compared the spatial patterns of the first five basic modes with those obtained from the whole population. (ii) Head motion. Head motion during R-fMRI scanning can affect the fluctuation amplitudes of BOLD signals[99]. Different from the main analysis, we used stricter head motion exclusion criteria for R-fMRI data in the HCP dataset (i.e., >2 mm or 2° in any direction or mean FD > 0.2 mm) to further reduce the influence of head motion. (iii) Global signal regression. In the main analysis, the global signal was regressed to better reduce the influence of head motion and non-neural signals[89,99]. To assess the potential influence of the global signal, we re-preprocessed the R-fMRI data in the HCP dataset without global signal regression. (iv) Brain parcellation. To assess the influence of spatial resolution, we extracted regional time courses from the HCP dataset by using the same type of functional parcellations with different spatial resolutions (i.e.,

comprising 200 and 400 cortical regions)[52]. The leading basic modes obtained from different spatial resolutions were compared at the functional system level[53] and the voxel-wise level. In the latter case, the voxels within the same nodal regions were assigned the same amplitude values for each basic mode, regardless of the spatial resolution. In cases (i)–(iv), the validation analysis was performed based on REST1 of the HCP dataset. (v) Reproducibility across datasets. We identified the leading basic modes from another independent dataset, i.e., the Beijing Zang dataset[51], and compared them with those in the HCP dataset.

**Statistics and reproducibility**. Permutation tests were performed to obtain the statistical significance of the following analyses, including the number of the leading basic modes (Fig. 1a), spatial similarities between the leading basic modes and cognitive components (Fig. 2b), and the influence of sleep deprivation on the fluctuation amplitudes and coactivation patterns of the leading basic modes (Fig. 4c, d). For each analysis, 10,000 permutation instances were generated during the permutation test. Paired $t$-tests were performed to test the statistical differences between intra- and inter-subject similarity of the reconstructed FC matrices between two runs ($n = 700$) (Fig. 3d). Five analysis strategies were considered to verify the reproducibility, including (i) varying the number of participants ($n$ ranged from 50 to 650 with a step of 50); (ii) using stricter head motion exclusion criteria ($n = 415$); (iii) performing nuisance regression without global signal regression ($n = 700$); (iv) defining brain nodes based on two functional parcellations with different spatial resolutions ($n = 700$); and (v) using another independent dataset, i.e., the Beijing Zang dataset ($n = 197$). The number and spatial patterns of the leading basis modes were examined in these cases.

**Reporting summary**. Further information on research design is available in the Nature Portfolio Reporting Summary linked to this article.

## Data availability

The S900 release of the HCP dataset is publicly available at https://www.humanconnectome.org/study/hcp-young-adult/data-releases. The Beijing Zang dataset used for the replication analysis is publicly available at https://www.nitrc.org/projects/fcon_1000. The sleep deprivation dataset is available upon reasonable request. Maps of leading basic modes and some other data supporting our results are available at https://github.com/liaolab-bnu/LeadingModes_rfMRI.

## Code availability

Codes used for data analysis are available at https://github.com/liaolab-bnu/LeadingModes_rfMRI.

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

## Acknowledgements

We thank Prof. Jingfang Fan and Prof. Dahui Wang for the valuable discussion. This work was supported by the National Natural Science Foundation of China (Nos. 81971690 and 11835003), the Tang Scholar Award, and the Fundamental Research Funds for Central Universities (No. 2019NTST24).

## Author contributions

X.C. and X.H.L. designed the research; Z.H.T., K.Z., L.Q.Z., and Z.T.Z collected parts of imaging dataset; X.C., H.D.R., X.H.C., X.S.C., Z.H.L., Y.H., and X.H.L. provided the methodological instruction; X.C. and X.H.L. performed the data analysis; X.C. and X.H.L. wrote the paper; X.C. and X.H.L. revised the paper.

## Competing interests

The authors declare no competing interests.
