## [Peer Review File · Communications Biology]

Reviewers' comments:

Reviewer #1 (Remarks to the Author):

I was slightly surprised to read that the authors ignored the rich literature surrounding time-varying network connectivity: e.g., <https://www.nature.com/articles/s41593-020-00719-y> and <https://www.ncbi.nlm.nih.gov/pmc/articles/PMC5073034/>, which is highly relevant to their approach.

The authors approach also shares similarities with the LEIDA approach: <https://sites.google.com/site/cvjoanacabral/codes/leida-leading-eigenvector-dynamics-analysis>. The authors should at the least comment on this, and most likely directly compare their results to this approach.

Given that the eigen-microstate analysis is novel to the field, I strongly recommend the authors include a summary/overview of the approach, along with its perceived strengths/weaknesses compared to other analytic methods, such as PCA, ICA, temporal functional modes and others? To my eye, the methods appear to be identical to PCA. If this is the case, why use a different name for something we already know and use? If it's not, then the authors need to say precisely how the methods differ, and how the ways in which they differ make a difference for interpretation. Unless we know how this method differs/overlaps with existing techniques, it is difficult to assess the novelty of figures 1 and 2, which are well-known relationships from other studies (e.g., <https://www.pnas.org/doi/10.1073/pnas.1608282113>).

Reviewer #2 (Remarks to the Author):

In the manuscript entitled "Leading Basic Modes of Spontaneous Activity Drive Individual Functional Connectivity Organization in the Resting Human Brain", the authors investigate the basic modes underpinning brain activity patterns using an eigen-microstate analysis based on SVD decomposition. By studying three resting-state functional MRI datasets, they discovered that a few dominant basic modes govern activity fluctuations, exhibiting specific spatial patterns that involve the separation of activity between the default-mode and primary attention regions.

Overall, the methods are sound, and the results are interesting. However, I believe that at this stage some additional clarifications and revisions are necessary before acceptance. Please find below a detailed assessment of the manuscript:

Main comments:

- The authors chose five basic modes based on the elbow criterion during the eigen-microstate analysis of 700 participants from the HCP dataset. It would be helpful to determine whether these five modes remain stable and robust even when analyzing a smaller subset of participants (e.g., 100 participants).

- The authors utilized the success rate metric introduced by Finn et al. in 2015 to evaluate the individual identification analysis. However, this metric is considered coarse as it can be viewed as a binary classification process. Perhaps, a more refined measure (Idiff), as the one proposed by Amico, E., & Goñi, J. (2018). *Scientific reports*, 8(1), 8254, might provide a more quantitative score for the

identification analysis according to the basic modes retained.

- The manuscript does not acknowledge other recent approaches for detecting brain states/modes that consider dynamic functional connectivity. The authors should include some of these references for the sake of completeness:

-Huntenburg, J. M., Bazin, P.-L. & Margulies, D. S. Large-scale gradients in human cortical organization. *Trends Cogn. Sci.* 22, 21–31 (2018)

-Cabral, Joana, et al. "Cognitive performance in healthy older adults relates to spontaneous switching between states of functional connectivity during rest." *Scientific reports* 7.1 (2017): 5135.

-de Alteriis, Giuseppe, et al. "EiDA: A lossless approach for the dynamic analysis of connectivity patterns in signals; application to resting state fMRI of a model of ageing." *bioRxiv* (2023): 2023-02.

-Casorso, Jeremy, et al. "Dynamic mode decomposition of resting-state and task fMRI." *Neuroimage* 194 (2019): 42-54.

Minor Comments:

- Lines 435-436: The description of the normalization of the nodal time course is equivalent to a z-score of the data across time. Perhaps, to make the explanation more concise, it might be preferable to use such terminology and add that, as a result, the nodal time course has zero mean and unit variance.

Reviewer #1:

1.1 I was slightly surprised to read that the authors ignored the rich literature surrounding time-varying network connectivity: e.g., <https://www.nature.com/articles/s41593-020-00719-y> and <https://www.ncbi.nlm.nih.gov/pmc/articles/PMC5073034/>, which is highly relevant to their approach.

R: We thank the reviewer for this suggestion. We have cited these references in the relevant paragraphs and briefly described them in the revised manuscript.

The relevant Introduction section (Line 13-18 and Line 33-35, Page 3):

“An innovative approach is to examine single frames of brain activity to reveal the transient coordination at shorter time scales (e.g., seconds) (Preti et al. 2017). The whole-brain activity patterns have been classified into several recurrent brain states with different coactivation patterns (Liu et al. 2013; Liu and Duyn 2013; Shine et al. 2016; Vidaurre et al. 2017). The temporal transition between these brain states follows a hierarchical structure (Vidaurre et al. 2017) and shows alterations across tasks (Bray et al. 2015; Chen et al. 2015; Shine et al. 2016), consciousness states (Di Perri et al. 2018; Huang et al. 2020), and psychiatric disorders (Zhuang et al. 2018; Yang et al. 2021).”

“Similarly, the edge-centric approach decomposes FC into framewise contributions (Faskowitz et al. 2020) and reveals dominant contributions of high-amplitude coactivations at critical time points (Zamani Esfahlani et al. 2020; Betzel et al. 2022).”

The relevant Method section (Line 27-31, Page 16):

“The original FC between two nodal regions is defined as the Pearson’s correlation between their time courses (Biswal et al. 1995). As each regional time course (A_{it} , $t = 1, \dots, M$) was normalized over time (i.e., mean = 0 and SD = 1), the FC between nodes i and j can be estimated as (Finn et al. 2015; Faskowitz et al. 2020):

$$FC_{ij} = \frac{1}{M-1} \sum_{t=1}^M A_{it} A_{jt}, \quad (5)$$

where M denotes the number of time points in the time course.”

1.2 The authors approach also shares similarities with the LEIDA approach:

<https://sites.google.com/site/cvjoanacabral/codes/leida-leading-eigenvector-dynamics-analysis>.

The authors should at the least comment on this, and most likely directly compare their results to this approach.

R: We appreciate the reviewer for pointing out this issue. We agree with the reviewer that there are similarities between the Leading Eigenvector Dynamics Analysis (LEiDA) approach and the eigen-microstate analysis, particularly in utilizing the leading eigenvector and the reconstruction of functional connectivity from multiple leading eigenvectors. However, these two approaches differ in their basic hypotheses and the interpretation of their results. The LEiDA analysis assumes that brain activity remains in a discrete brain state at each time, and each brain state corresponds to a state with a unique functional connectivity pattern (Cabral et al. 2017). The

leading eigenvector is identified at each time point to capture the dominant functional connectivity pattern specific to that time point. On the other hand, the eigen-microstate analysis assumes that brain activity may exhibit commonalities over time, with instantaneous brain activity arising from the combination of multiple basic modes (Hu et al. 2019; Sun et al. 2021). Each leading basic mode, or eigen-microstate, is identified by a weighted sum of activity patterns at all time points, representing a typical activity pattern underlying the rich repertoire of brain activity. Thus, the LEiDA approach and the eigen-microstate analysis explore the spatiotemporal organization of brain function from different perspectives: interregional connectivity and regional activity, respectively.

In the study by Cabral et al. (2017), the Leading Eigenvector Dynamics Analysis (LEiDA) approach identified typical connectivity modes in the healthy older adults, such as one mode with global coherence across the brain and one mode with high coherence within the default-mode network. These two modes align with the leading basic modes in our study when the global signal is retained. These spatial similarities between the two approaches further support the vital contribution of the leading basic activity modes to dynamic functional connectivity patterns. The discrepancies observed in other modes across studies may be attributed to differences in brain coverage, node definitions, and the specific populations and features of interest examined in each study. In the revised manuscript, we have added a brief comparison with this approach in the Discussion section.

The relevant Discussion section (Line 13-28, Page 10):

“The leading basic modes identified here show intriguing spatial similarities with dominant connectivity patterns or modes reported in previous studies (Margulies et al. 2016; Cabral et al. 2017; Casorso et al. 2019). Specifically, the first three leading basic modes exhibit consistent patterns with the first three gradients of the cortical FC pattern previously identified (Margulies et al. 2016). Similarly, the Leading Eigenvector Dynamics Analysis (LEiDA) approach identified typical connectivity modes in healthy older adults (Cabral et al. 2017), such as one mode with global coherence across the brain and one mode with high coherence within the default-mode network. These two modes align with the leading basic modes in our study when the global signal is retained. Moreover, the dynamic mode decomposition identified typical FC modes with different temporal features (e.g., damping time and oscillatory periods) (Casorso et al. 2019). The first dynamic mode, characterized by an anti-correlation between the default-mode and task-positive networks, partially overlaps with the second leading basic mode in our study. Similar patterns between different types of maps further support the notion that leading basic modes play a dominant role in shaping the FC pattern. The discrepancies observed in other modes across studies may be attributed to differences in brain coverage, node definitions, and specific populations and features of interest examined in each study. These methodological variations should be considered when interpreting and comparing results across studies.”

1.3 Given that the eigen-microstate analysis is novel to the field, I strongly recommend the authors include a summary/overview of the approach, along with its perceived strengths/weaknesses compared to other analytic methods, such as PCA, ICA, temporal functional modes and others? To my eye, the methods appear to be identical to PCA. If this is the case, why use a different name for something we already know and use? If it's not, then the authors need to say precisely how the methods differ, and how the ways in which they differ make a difference for interpretation. Unless we know how this method differs/overlaps with

existing techniques, it is difficult to assess the novelty of figures 1 and 2, which are well-known relationships from other studies (e.g., <https://www.pnas.org/doi/10.1073/pnas.1608282113>).

R: We appreciate the reviewer's valuable suggestions. The eigen-microstate analysis is an approach specifically designed to extract basic activity modes (i.e., eigen-microstates) from the temporal evolution of a system. In our study, we utilized this approach to identify basic modes as a weighted sum of activity patterns at all time points. The leading basic modes exhibit spatial similarities with functional connectivity gradients reported in Margulies et al. (2016), but they reflect representative activity patterns rather than interregional relationships. By exploring spatial associations of these leading basic modes with brain activation maps and cognitive components, our study provides a direct comparison of activity patterns between resting and task states, enriching the knowledge regarding the functional significance of spontaneous activity. In the revised manuscript, we have included a brief discussion regarding the relationship with Margulies et al. (2016). The differences with other approaches are clarified as follows.

Differences with PCA. While the estimation of eigen-microstates shares a similar formula with the principal component analysis (PCA), the eigen-microstate analysis and PCA are used in different contexts. The eigen-microstate analysis focuses on the temporal evolution of a system. It aims to identify typical spatial patterns (i.e., eigen-microstates) of brain activity by analyzing the relationship between time points. On the other hand, PCA is commonly used to reduce the dimensionality of a dataset by finding the principal components that capture the main variance in the multivariate data. PCA emphasizes the relationship between variables of data (here, brain regions). Moreover, unlike PCA, which only captures relative weights of variables, the eigen-microstate analysis provides an intuitive biological interpretation of eigen-microstates as typical activity states. We also demonstrated that each eigen-microstate corresponds to a specific connectivity pattern through further theoretical modeling, which is not covered by PCA analysis. Therefore, the application of the eigen-microstate analysis can provide intuitive biological interpretability of basic modes and establish their relationship with connectivity patterns, offering a new way to explore functional organization principles of spontaneous activity.

Differences with ICA and temporal functional modes. Independent component analysis (Calhoun et al. 2009) and temporal functional mode analysis (Smith et al. 2012) are used to find spatial components that are spatially independent and temporally independent, respectively. These approaches primarily focus on interregional functional homogeneity. The identified components are often referred to as functional networks and each spatial pattern reflects a set of weights over brain nodes. In contrast, the eigen-microstate analysis focuses on organization principles regarding spatial patterns of brain activity *per se*. The identified leading basic modes reflect typical activity patterns that coexist over time, allowing for an intuitive biological interpretation. Thus, the eigen-microstate analysis and these other approaches decompose the temporal evolution of brain activity from different perspectives, depending on the specific questions of interest.

In the revised version, we have rephrased paragraphs regarding the eigen-microstate analysis to highlight its core concept. Meanwhile, we have also added some comparisons with other approaches in the Discussion section.

The relevant Introduction section (Line 38, P3 and Line 1-4, Page 4):

“To address these issues, we leveraged a novel statistical physics approach, i.e., the eigen-

microstate analysis (Hu et al. 2019; Sun et al. 2021), to identify basic modes of spontaneous activity of the resting human brain. The eigen-microstate analysis is useful for extracting meaningful and fundamental spatial components (i.e., basic modes) underlying the temporal evolution of complex systems by incorporating spatial information over time (Hu et al. 2019; Sun et al. 2021)”

The relevant Methods section (Page 13-15):

“We applied a novel eigen-microstate analysis (Hu et al. 2019; Sun et al. 2021) from the statistical physics theory to the HCP dataset to identify basic modes underlying the rich repertoire of brain activity. The eigen-microstate analysis is an approach specifically designed to extract basic activity modes (i.e., eigen-microstates) of the temporal evolution of a system. This approach has been applied to several complex systems (e.g., the Earth system and stock markets) and reveals meaningful activity modes that show well-defined spatial patterns (Sun et al. 2021). In this study, the eigen-microstate analysis was applied to time courses of each R-fMRI run (i.e., REST1 and REST2) separately. The brain microstates were defined at the nodal level to reduce computational burden.

Definition of the ensemble matrix and microstates. First, we defined 1000 cortical nodes based on a prior functional parcellation (Schaefer et al. 2018), which would enhance functional homogeneity within each nodal region. Then, we extracted the time courses of these nodes for each participant. The time course for each node was further transformed into z-score values with zero mean and unit variance over time. Finally, the normalized nodal time courses were concatenated across participants, resulting in an $N \times M$ time course matrix \mathbf{A} , where N denotes the number of nodes (i.e., 1000 here) and M denotes the number of time points in the concatenated time courses (i.e., 1185×700). Matrix \mathbf{A} was considered as an ensemble matrix representing a rich repertoire of brain activity, and each column, \mathbf{A}_m , represents a microstate of brain activity at a specific time point from a statistical physics perspective.

Theoretical derivation of eigen-microstates. The eigen-microstate analysis assumes that microstates may exhibit commonalities over time, with instantaneous brain activity arising from the combination of multiple basic modes. It aims to identify the basic modes (i.e., eigen-microstates) that represent fundamental building blocks of rich microstates and are independent of each other (Hu et al. 2019; Sun et al. 2021). Specifically, we first calculated the covariance matrix between microstates as $\mathbf{C} = \mathbf{A}^T \mathbf{A}$ (Sun et al. 2021), which reflects the spatial similarity between all microstates. Next, we computed the eigenvectors of the matrix \mathbf{C} ,

$$\mathbf{C} \mathbf{v}_i = \lambda_i \mathbf{v}_i, \quad (1)$$

wherein \mathbf{v}_i is the i th eigenvector of the matrix \mathbf{C} , and λ_i is the corresponding eigenvalue. The representativeness of the spatial patterns of microstates at different time points is captured by the eigenvectors (Bonacich 2007), which take into account the entire spatial similarity pattern between all microstates. Finally, we obtained each eigen-microstate as the weighted sum of the origin microstates (Hu et al. 2019; Sun et al. 2021):

$$\mathbf{E}_i = \mathbf{A} \mathbf{v}_i = \sum_{t=1}^M A_t v_{ti}, \quad (2)$$

wherein v_{ti} denotes the t th element of eigenvector \mathbf{v}_i and quantifies the contribution of t th microstate to the i th eigenvector. In other words, the basic modes (i.e., eigen-microstates) were

identified by incorporating brain activity patterns over time and thus can be intuitively interpreted as typical activity patterns. The weight factor of a given eigen-microstate \mathbf{E}_i is defined as $w_i = \mathbf{E}_i^T \mathbf{E}_i = \lambda_i$.

Calculation of eigen-microstates. In practical analysis, it is difficult to derive the eigenvectors of the covariance matrix \mathbf{C} one by one due to the large number of time points M . Therefore, we obtained the eigen-microstates with the aid of the singular value decomposition (SVD) (Sun et al. 2021). Based on the SVD, the ensemble matrix $\mathbf{A}_{N \times M}$ was factorized as the product of three matrices:

$$\mathbf{A}_{N \times M} = \mathbf{U}_{N \times N} \mathbf{\Sigma}_{N \times M} \mathbf{V}_{M \times M}^T, \quad (3)$$

wherein $\mathbf{U} = [\mathbf{u}_1, \mathbf{u}_2, \dots, \mathbf{u}_N]$ and $\mathbf{V} = [\mathbf{v}_1, \mathbf{v}_2, \dots, \mathbf{v}_M]$ contain eigenvectors for two matrices of $\mathbf{A}\mathbf{A}^T$ and $\mathbf{A}^T\mathbf{A}$, respectively, and $\mathbf{\Sigma}_{N \times M}$ is a diagonal matrix of singular values (σ_i). By substituting this decomposition into Eq. (2), the definition of the eigen-microstate can be reformulated as:

$$\mathbf{E}_i = \mathbf{A}\mathbf{v}_i = \sigma_i \mathbf{u}_i, \quad (4)$$

wherein \mathbf{u}_i is the eigenvector of the matrix $\mathbf{A}\mathbf{A}^T$. The weight of each eigen-microstate can be obtained as $w_i = \mathbf{E}_i^T \mathbf{E}_i = \sigma_i^2$. The matrix \mathbf{A} was normalized by dividing the root-sum-square of all its elements (i.e., a dataset-dependent constant S) before SVD to ensure $\sum \sigma_i^2 = 1$.

Of note, the estimation for eigen-microstates in Eq. (4) is similar to the principal component analysis (PCA) (Hastie et al. 2009). However, the eigen-microstate analysis and PCA are used in different contexts. The former is specific to the temporal evolution of a system and is used to identify the typical spatial patterns of microstates by analyzing the spatial relationship between time points (see Eq. (2)). The latter, however, is a technique commonly used to reduce the dimensionality of a dataset by finding the principal components that capture the main variance in the multivariate data. In this sense, PCA focuses on the relationship between data variables (here, brain regions) rather than the relationship between time points. Moreover, compared to PCA, which only captures the relative weights of variables, eigen-microstate analysis can provide an intuitive biological interpretation of the eigen-microstates as typical activity states (see Eq. (2)).”

The relevant Discussion section (Line 17-31 Page 8):

“We note that the eigen microstate analysis shares similarities with several existing methods that are used to decode the spatiotemporal organization of spontaneous brain activity. For example, several component analysis methods, such as the principal component analysis (PCA) (Zhong et al. 2009), independent component analysis (ICA) (Calhoun et al. 2009), and temporal functional mode analysis (Smith et al. 2012), have been applied to fMRI time series to identify dominant interregional interaction patterns. The specified spatial components are often regarded as functional networks (Smith et al. 2009; Zhong et al. 2009; Calhoun and Adali 2012), while their spatial patterns reflect the relative weights of brain regions. Some other relevant approaches have detected brain states or modes by considering dynamic functional connectivity patterns (Allen et al. 2014; Shine et al. 2016; Cabral et al. 2017; Betzel et al. 2022; de Alteriis et al. 2023). Each connectivity state differs from the other in terms of the overall connectivity pattern (Allen et al. 2014) or dominant connectivity modes quantified by leading eigenvectors (Cabral et al. 2017; Casorso et al. 2019). Different from these previous approaches, the eigen-microstate analysis focuses primarily on spatial patterns of instantaneous activity *per se* and further establishes a bridge between instantaneous brain activity and functional connectivity patterns. Moreover, the

eigen-microstate analysis assumes that multiple basic modes may coexist over time rather than a dominant brain state at each time, which allows for capturing delicate changes in brain activity over time.”

Reviewer #2:

In the manuscript entitled "Leading Basic Modes of Spontaneous Activity Drive Individual Functional Connectivity Organization in the Resting Human Brain", the authors investigate the basic modes underpinning brain activity patterns using an eigen-microstate analysis based on SVD decomposition.

By studying three resting-state functional MRI datasets, they discovered that a few dominant basic modes govern activity fluctuations, exhibiting specific spatial patterns that involve the separation of activity between the default-mode and primary attention regions.

Overall, the methods are sound, and the results are interesting. However, I believe that at this stage some additional clarifications and revisions are necessary before acceptance. Please find below a detailed assessment of the manuscript.

R: We thank the reviewer for the positive comments and valuable suggestions.

2.1 The authors chose five basic modes based on the elbow criterion during the eigen-microstate analysis of 700 participants from the HCP dataset. It would be helpful to determine whether these five modes remain stable and robust even when analyzing a smaller subset of participants (e.g., 100 participants).

R: In the revised version, we have performed additional analysis to validate the potential influence of the number of participants on the identification of the leading basic modes. Specifically, we re-performed the eigen-microstate analysis by including R-fMRI data from different numbers of participants, with the participant number varying from 50 to 650 with a step of 50. Given a participant number, participants were randomly sampled from the whole population of the HCP dataset and this process was repeated 100 times. Using REST1 of the HCP dataset, we observed that the number of the leading basic modes was fixed at five when the participant number was equal to or greater than 250. Their spatial patterns were highly similar to those obtained from the whole population (i.e., 700 participants) with one-to-one correspondence (all mean $r_s > 0.96$). In the revised version, we have added the relevant analyses and results in the Methods and Results sections.

The relevant Results section (Line 26-35 Page 7):

“We further assessed the reliability of the presence and spatial patterns of the leading basic modes (Figs. S8-S13). Five additional analysis strategies were considered, including (i) using different numbers of participants; The number of the leading basic modes reached five without variation when the participant number considered was equal to or greater than 250 (Fig. S8). Meanwhile, spatial patterns of these leading basic modes were highly similar to those obtained from the whole population (i.e., 700 participants) with a one-to-one correspondence (all mean $r_s > 0.96$, Fig. S8).”

The relevant Methods section (Line 23-31 Page 18; Line 6-7 Page 19):

“The reliability of the leading basic modes was validated by considering five analysis strategies that may affect the identification of the leading basic modes. In each case, the leading basic modes were re-estimated and compared with those obtained in the main analyses (i.e., REST1 in HCP). (i) Number of participants. We re-identified the leading basic modes by including R-fMRI data from different numbers of participants from the HCP dataset, with the number of

participants varying from 50 to 650 with a step of 50. Given a participant number, we randomly sampled participants from all the 700 participants, and this process was repeated 100 times. For each sampling instance, we counted the number of the leading basic modes and compared the spatial patterns of the first five basic modes with those obtained from the whole population.In cases (i) - (iv), the validation analysis was performed based on REST1 of the HCP dataset.”

The relevant Supplement section (Page 8):

“**Figure S8.** Influence of the number of participants on the identification of leading basic modes (REST1 of the HCP dataset). (A) Dependence of the number of the leading basic modes on the number of participants. Given a participant number, we randomly sampled participants from the whole population (i.e., 700 participants) and re-identified the leading basic modes. This process was repeated 100 times to obtain stable results. Given a participant number, we display the mean and standard deviation of the number of the leading basic modes across 100 sampling instances. We found that the presence of five leading basic modes was robust without variance when the number of participants was equal to or greater than 250. (B) Spatial similarity of the first five basic modes with those obtained from the whole population. Here we display the mean and standard deviation of spatial similarity for each basic mode, separately. We found that for each basic mode the spatial similarity was high with the corresponding mode in the main results and reached a saturation point after 250”

2.2 The authors utilized the success rate metric introduced by Finn et al. in 2015 to evaluate the individual identification analysis. However, this metric is considered coarse as it can be viewed as a binary classification process. Perhaps, a more refined measure (I_{diff}), as the one proposed by Amico, E., & Goñi, J. (2018). Scientific reports, 8(1), 8254, might provide a more quantitative score for the identification analysis according to the basic modes retained.

R: Thank the reviewer for this valuable suggestion. In the revised manuscript, we have added the I_{diff} measure into the identification analysis to further quantify the individual identifiability of the connectivity matrix. Similar with the identification accuracy, we found that I_{diff} of the reconstructed matrix also increased with increasing numbers of basic modes. Of note, the I_{diff}

value exceeded that obtained from the original functional connectivity matrix when the first five basic modes were included, suggesting it contains more individualized information than the original functional connectivity matrix. We have updated the relevant Methods and Results sections, and Figure 2 in the revised version.

The relevant Results section (Line 22-30 Page 6 and Page 28):

“Then, we performed the individual identification analysis (Finn et al. 2015) by comparing individual FC matrices between two runs. We observed an identification accuracy of 97% based on the original FC matrix. For reconstructed individual FC matrices, the identification accuracy increased rapidly with increasing number of basic modes and reached 92% when the five leading basic modes were included (Fig. 3E). Similarly, the differential identifiability I_{diff} , which quantifies the difference between the mean intra-subject similarity and the mean inter-subject similarity (Amico and Goñi, 2018), also increased rapidly with increasing numbers of basic modes. Its value reached 30.3% when all the five leading basic modes were included, which exceeded I_{diff} of the original FC matrix (Fig. 3E). These results suggest that these leading basic modes make the dominant contribution to the individualized functional organization.”

“**Figure 3.** Reconstructing functional connectivity based on the basic modes. (A) Spatial similarity between the reconstructed and original FC matrices at the population level (REST1). The original FC matrix was estimated based on the concatenated time courses of all participants. The reconstructed FC matrix was generated separately by using different numbers of basic modes based on the theoretical model. M denotes the number of time points, N denotes the number of all possible basic modes, and E_{ik} is the i th element of the k th basic mode. (B)

Reconstructing the FC matrix with the first five basic modes (REST1). Left, spatial patterns of the reconstructed and original FC matrices; right, spatial similarity between these two matrices. (C) Spatial similarity between the reconstructed and original FC matrices at the individual level for both runs. The reconstructed FC matrix was generated by using different numbers of basic modes. Mean spatial similarity across participants and the corresponding standard deviation are displayed. The distributions of individual spatial similarity obtained from the five leading basic modes are shown in the subplot in the form of the violin plots and the box plots. (D) Intra- and inter-subject similarity of the reconstructed FC matrices between two runs. Mean similarity across participants and the corresponding standard deviation are displayed. *, denotes significant differences (paired *t*-tests, *ps* < 0.05). (E) Individual identification accuracy and differential identifiability I_{diff} based on the reconstructed FC matrix. Individual FC matrices were reconstructed with different numbers of basic modes. The dashed lines denote the identification accuracy and I_{diff} based on the original FC matrix. FC, functional connectivity; DMN, default-mode network; FPN, frontoparietal network; LN, limbic network; VAN, ventral attention network; DAN, dorsal attention network; SMN, somatomotor network; VN, visual network.”

The relevant Methods section (Line 29-33 Page 17):

“We also evaluated the differential identifiability (Amico and Goñi 2018) to quantify the individual uniqueness of the reconstructed FC matrices. The differential identifiability, I_{diff} , was defined as $I_{diff} = (I_{self} - I_{others}) \times 100$, wherein I_{self} and I_{others} denoted the mean intra-subject similarity and mean inter-subject similarity between two runs, respectively. A larger I_{diff} value indicates more individual-specific information is captured.”

The relevant Discussion section (Lined 31-38 Page 10)

“Notably, the FC pattern reconstructed by only the first leading basic mode was highly similar to the original pattern but showed a moderate performance in individual identification and a relative low differential identifiability value. Identification accuracy and differential identifiability increased rapidly when subsequent basic modes were included, indicating that more individual-specific information is captured by subtly modulating the backbone of the FC pattern.”

2.3 The manuscript does not acknowledge other recent approaches for detecting brain states/modes that consider dynamic functional connectivity. The authors should include some of these references for the sake of completeness:

-Huntenburg, J. M., Bazin, P.-L. & Margulies, D. S. Large-scale gradients in human cortical organization. *Trends Cogn. Sci.* 22, 21 – 31 (2018)

-Cabral, Joana, et al. "Cognitive performance in healthy older adults relates to spontaneous switching between states of functional connectivity during rest." *Scientific reports* 7.1 (2017): 5135.

-de Alteriis, Giuseppe, et al. "EiDA: A lossless approach for the dynamic analysis of connectivity patterns in signals; application to resting state fMRI of a model of ageing." *bioRxiv* (2023): 2023-02.

-Casorso, Jeremy, et al. "Dynamic mode decomposition of resting-state and task fMRI." *Neuroimage* 194 (2019): 42-54.

R: We thank the reviewer for this valuable information. We have cited these references in the relevant paragraphs and added a brief discussion regarding these approaches in the Discussion

section.

The relevant Discussion section (Line 17-31 Page 8; Line 26-33 Page 9; Line 13-28 Page 10):

“We note that the eigen microstate analysis shares similarities with several existing methods that are used to decode the spatiotemporal organization of spontaneous brain activity. For example, several component analysis methods, such as the principal component analysis (PCA) (Zhong et al. 2009), independent component analysis (ICA) (Calhoun et al. 2009), and temporal functional mode analysis (Smith et al. 2012), have been applied to fMRI time series to identify dominant interregional interaction patterns. The specified spatial components are often regarded as functional networks (Smith et al. 2009; Zhong et al. 2009; Calhoun and Adali 2012), while their spatial patterns reflect the relative weights of brain regions. Some other relevant approaches have detected brain states or modes by considering dynamic functional connectivity patterns (Allen et al. 2014; Shine et al. 2016; Cabral et al. 2017; Betzel et al. 2022; de Alteriis et al. 2023). Each connectivity state differs from the other in terms of the overall connectivity pattern (Allen et al. 2014) or dominant connectivity modes quantified by leading eigenvectors (Cabral et al. 2017; Casorso et al. 2019). Different from these previous approaches, the eigen-microstate analysis focuses primarily on spatial patterns of instantaneous activity per se and further establishes a bridge between instantaneous brain activity and functional connectivity patterns. Moreover, the eigen-microstate analysis assumes that multiple basic modes may coexist over time rather than a dominant brain state at each time, which allows for capturing delicate changes in brain activity over time.”

“The presence of the leading basic modes in intrinsic activity might be shaped by anatomical substrates of the brain, given the tight structure-function coupling of the brain (Park and Friston 2013; Suarez et al. 2020). Previous studies have demonstrated that spatial arrangements of cortical microstructures show a dominant gradient spanning between sensorimotor-to-transmodal areas (Huntenburg et al. 2018). For example, the myelination map of the brain (Glasser et al. 2016) shows spatial similarities with the first three leading basic modes observed here, indicating a potential link between the macroscale brain activity and the local microstructure. However, how these leading basic modes emerge from the anatomical properties, such as myelination, cortical thickness, and white-matter connectivity, requires further investigation”

“The leading basic modes identified here show intriguing spatial similarities with dominant connectivity patterns or modes reported in previous studies (Margulies et al. 2016; Cabral et al. 2017; Casorso et al. 2019). Specifically, the first three leading basic modes exhibit consistent patterns with the first three gradients of the cortical FC pattern previously identified (Margulies et al. 2016). Similarly, the Leading Eigenvector Dynamics Analysis (LEiDA) approach identified typical connectivity modes in healthy older adults (Cabral et al. 2017), such as one mode with global coherence across the brain and one mode with high coherence within the default-mode network. These two modes align with the leading basic modes in our study when the global signal is retained. Moreover, the dynamic mode decomposition identified typical FC modes with different temporal features (e.g., damping time and oscillatory periods) (Casorso et al. 2019). The first dynamic mode, characterized by an anti-correlation between the default-mode and task-positive networks, partially overlaps with the second leading basic mode in our study. Similar patterns between different types of maps further support the notion that leading basic modes play a dominant role in shaping the FC pattern. The discrepancies observed in other modes across studies may be attributed to differences in brain coverage, node definitions, and specific

populations and features of interest examined in each study. These methodological variations should be considered when interpreting and comparing results across studies.”

2.4 Lines 435-436: The description of the normalization of the nodal time course is equivalent to a z-score of the data across time. Perhaps, to make the explanation more concise, it might be preferable to use such terminology and add that, as a result, the nodal time course has zero mean and unit variance.

R: Done.

The relevant Method section (Line 9-11 Page 14):

“Then, we extracted the time courses of these nodes for each participant. The time course for each node was further transformed into z-score values with zero mean and unit variance over time.”

REVIEWERS' COMMENTS:

Reviewer #1 (Remarks to the Author):

The authors have addressed my concerns.

Reviewer #3 (Remarks to the Author):

The authors have successfully addressed all of my concerns. The revised version of the manuscript shows significant improvement compared to the previous version. The main manuscript and methods sections now contain the necessary information to comprehend the work they have conducted. As such, I have no further comments on this manuscript version.